# WHOQOL-BREF in Measuring Quality of Life Among Sickle Cell Disease Patients with Leg Ulcers

**DOI:** 10.3390/ijerph22010108

**Published:** 2025-01-15

**Authors:** Caroline Conceição da Guarda, Jéssica Eutímio de Carvalho Silva, Gabriela Imbassahy Valentim Melo, Paulo Vinícius Bispo Santana, Juliana Almeida Pacheco, Bruno Terra Correa, Edvan do Carmo Santos, Elisângela Vitória Adorno, Andrea Spier, Teresa Cristina Cardoso Fonseca, Marilda Souza Goncalves, Milena Magalhães Aleluia

**Affiliations:** 1Laboratório de Patologia Aplicada e Genética, Departamento de Ciências Biológicas, Universidade Estadual de Santa Cruz, Ilhéus 45662-900, BA, Brazil; cguarda4@hotmail.com (C.C.d.G.); jessicaeutimio@gmail.com (J.E.d.C.S.); gabriela_melo00@hotmail.com (G.I.V.M.); vinibsantana13@gmail.com (P.V.B.S.); enfajulianna@gmail.com (J.A.P.); terrabruno@hotmail.com (B.T.C.); edvanrih@gmail.com (E.d.C.S.); 2Laboratório de Pesquisa em Anemias, Departamento de Análises Clínicas e Toxicológicas, Faculdade de Farmácia, Universidade Federal da Bahia, Salvador 40170-115, BA, Brazil; liuadorno@hotmail.com; 3Centro de Referência em Doença Falciforme, Secretaria Municipal de Saúde de Itabuna, Itabuna 45600-075, BA, Brazil; nutrirmeas@gmail.com; 4Departamento de Ciências da Saúde, Universidade Estadual de Santa Cruz, Ilhéus 45662-900, BA, Brazil; teresacrfonseca@gmail.com; 5Laboratório de Investigação em Genética e Hematologia Translacional, Instituto Gonçalo Moniz, Fundação Oswaldo Cruz, Salvador 40296-710, BA, Brazil; marilda.goncalves@fiocruz.br

**Keywords:** sickle cell disease, sickle cell leg ulcer, quality of life, WHOQOL-BREF

## Abstract

Sickle cell disease (SCD) presents complex clinical manifestations influenced by genetic, social, environmental, and healthcare access factors as well as socioeconomic status. In this context, sickle cell leg ulcers (SLUs) are a debilitating complication of SCD. We aimed to describe sociodemographic data and evaluate the quality of life (QoL) of SCD patients with and without SLUs. We conducted a cross-sectional study including 13 SCD patients with SLUs and 42 without LUs. Clinical data were obtained by reviewing the medical records, and QoL was assessed with the WHOQOL-BREF questionnaire. Our cohort of patients had a mean age of 34.9 years, with 52.8% male, 52.8% identifying as black, and 41.7% identifying as brown. Most had low income, incomplete education, and high unemployment rates. The social habits and relationships of SCD patients showed varying levels of friendship and family closeness, and the majority of SLU+ patients did not practice sports. We failed to find statistical differences in the WHOQOL-BREF domains between SLU+ and SLU− patients. However, higher income and employment status were associated with improved WHOQOL-BREF domain scores in SCD patients, while vaso-occlusive episodes and female gender were linked to lower scores. Our data reinforce the sociodemographic characteristics of SCD. The physical domain was associated with income, occupation, and vaso-occlusion. The psychological domain was associated with income and occupation. The social relationship domain was associated with occupation and female gender. The environmental domain was associated with vaso-occlusion. The WHOQOL-BREF is a reliable tool to measure QoL in SCD.

## 1. Introduction

Sickle cell disease (SCD) is a genetic hereditary disorder resulting from a mutation in the sixth codon of the β globin gene, leading to the emergence of hemoglobin S (HbS) [1,2]. Homozygosity for this allele constitutes sickle cell anemia (SCA, HbSS), and heterozygosity with another hemoglobin variant, such as hemoglobin C (HbC), results in SC hemoglobinopathy (HbSC) [3,4]. There is a wide range of clinical manifestations associated with SCD, one of which is the occurrence of sickle cell leg ulcers (SLUs) [5,6].

Although the pathophysiology of sickle cell leg ulcers (SLUs) is not fully elucidated, a hemolytic mechanism resulting from HbS polymerization, which leads to the occlusion of blood vessels and low tissue oxygenation, is frequently associated with these lesions [7,8,9], which mainly affect the lateral and medial malleoli of the ankle, unilaterally or bilaterally, due to favorable blood circulation, low amount of adipose tissue, and higher susceptibility to mechanical impact.

Our group has previously identified molecular biomarkers associated with SLUs and biologically related to hemolysis [10]; in addition, we found that hematological parameters were associated with clinical characteristics of SLUs [11]. Considered a painful and debilitating condition that can be acute or chronic, SLUs can have significant physical, psychological, and social impacts, significantly affecting the health-related quality of life (HRQoL) of these patients [12,13].

The assessment of quality of life (QoL) is considered imperative in all medical specialties and has been strongly recommended in routine healthcare practice [14,15,16]. In recent years, self-reported or patient-reported outcomes and QoL features related to physical and psychosocial health have been used to assess overall well-being [17]. Therefore, it is essential to understand the specific domains in which HRQoL appears to be reduced in SCD patients. However, much of the literature on the disease focuses on healthcare utilization and pain management. Robust evidence concerning the measurement of HRQoL over time is still lacking [14].

The WHOQOL-BREF is a tool developed by the World Health Organization (WHO) and is a globally applicable instrument for assessing QoL. It spans various areas of healthcare across different cultural contexts, languages, and countries. This self-assessment instrument addresses four dimensions of QoL, namely, “Physical Health”, “Psychological Health”, “Social Relationships”, and “Environment”, and includes two items examined separately: the “overall perception of QoL” and the “overall perception of health” of the individual. A study published in 2021 validated the psychometric properties [5,6] of the WHOQOL-BREF for evaluating SCD patients, demonstrating its effectiveness in assessing the QoL of these patients. Therefore, this assessment tool emerges as a highly recommended resource for monitoring and improving the QoL in this patient group.

Thus, the present study aimed to describe the sociodemographic characteristics of SCD patients with SLUs, as well as their QoL. In addition, we described the associations between WHOQOL-BREF scores and patients’ characteristics.

## 2. Materials and Methods

### 2.1. Study Design and Patients

The present cross-sectional study was conducted between October 2021 and October 2022 in the southern region of Bahia, Brazil. We included 55 participants diagnosed with SCD (HbSS and HbSC) who were assisted at the Centro de Referência de Doença Falciforme de Itabuna (CERDOFI), Bahia, Brazil; the Itabuna Sickle Cell Disease Reference Center (Programa de Doença Falciforme de Ilhéus, Centro de Referência de Doença Falciforme de Itabuna, CERDOFI); and the Ilhéus Sickle Cell Disease Program (Programa de Doença Falciforme de Ilhéus (PRODOFI), Bahia, Brazil. The study sample comprised children, adolescents, and adults of both genders, aged 16 to 57 years, of whom 13 participants had SLUs (SLU+). As a control group, we selected 42 participants who had a diagnosis of SCD in the same period according to the study’s inclusion criteria but did not have an active sickle cell leg ulcer at the time of the study (SLU−).

### 2.2. Inclusion and Exclusion Criteria

The inclusion criteria were for patients aged 12 years and older with SCD (HbSS and HbSC) who were seen at CERDOFI and PRODOFI during the study period. Participants who did not present HbSS or HbSC genotypes, those who had received transfusion therapy three months before the study, and those who did not consent to participate were excluded from the study.

### 2.3. Ethical Requirements

Participation in the study required the agreement and consent of the participants by signing the Informed Consent Form (ICF), which was obtained from participants older than 18 years, as well as the legal guardians of underage participants. The protocol duly approved by the Ethics and Research Committee of the Universidade Estadual de Santa Cruz (UESC) through protocol No. 47456021.4.0000.5526. The study was carried out according to the ethical principles established by the Declaration of Helsinki (1964) and its subsequent amendments.

### 2.4. Clinical, Sociodemographic, and Socioeconomic Data

Data collection was performed through the evaluation of the medical records and interviews with the application of a standardized questionnaire to assess the clinical–epidemiological, sociodemographic, and socioeconomic profile. All the patients who were followed up at the reference centers had regular appointments with psychologists and social services. Regarding drug therapy, all patients took folic acid and hydroxyurea. Patients with lesions underwent a clinical assessment of the lesion, hygiene, dressing, and debridement. Some SLU+ individuals used topical treatments with curative substances, and some used a combination of more than one substance. Hydrogels, bioactive dressings, collagenase, and the Unna boot were among the substances and dressings used.

### 2.5. Quality-of-Life Data

To evaluate the quality of life, we used the WHOQOL-BREF (World Health Organization of Quality of Life) tool developed by the World Health Organization (WHO). The WHOQOL-BREF has shown good to excellent reliability and validity and has four domains: physical, psychological, social, and environmental. This questionnaire is a shortened version of the WHOQOL-100. It consists of 26 questions that analyze 24 facets in four domains (D1: Physical, D2: Psychological, D3: Social relationship, and D4: Environment) so that it values individual perception and aims to assess the quality of life through answers that include intensity (“not at all” to “extremely”), capacity (“not at all” to “completely”), frequency (“never” to “always”), and evaluation (“very dissatisfied” to “very satisfied”; “very bad” to “very good”). The mean score of items within each domain is used to calculate the domain score. The results in a domain consist of the sum of the results for its items. A higher sum of points represents a higher quality of life in a single domain [17].

### 2.6. Statistical Analysis

Statistical analyses were developed using the Statistical Package for the Social Sciences (SPSS) version 20.0 software (IBM, Armonk, NY, USA). The WHOQOL-BREF domains of the study participants are expressed as means and respective standard deviations as well as medians and interquartile ranges. Each characteristic’s frequency and feeding profile are described as the absolute count and relative percentage. The reliability of each domain was calculated using Cronbach’s alpha coefficient. The distribution of each variable was tested by employing the Shapiro–Wilk test. The Mann–Whitney U test was used to compare groups. Fisher’s exact test was used to compare categorical variables. *p*-values < 0.05 were considered statistically significant.

## 3. Results

Sociodemographic characterization of SCD patients revealed that our cohort was aged 34.9 ± 10.7 years and consisted of 52.8% males; 14.5% were diagnosed with SCD before 6 months of age, 49% received the diagnosis from 6 months to 17 years, and 36.5% were diagnosed when they were at least 18 years old. Concerning ethnicity, 52.8% declared themselves to be black, and 41.7% reported themselves to be brown. The income per capita was low: 25.5% of the patients received less than the minimum wage, and 54.5% received only the minimum wage (Table 1).

When the interview was conducted, 35.8% of the patients had completed middle school or had started but not finished high school, and 43.3% had completed high school or had started but not finished a college degree. Furthermore, 49% had siblings with SCD, whereas 31.1% did not have relatives with SCD. Among the patients, 60% were taking hydroxyurea. Only 36.4% had an occupation, and 63.6% were unemployed. Overall, 72.5% did not consume alcoholic beverages, 96.1% were non-smokers, and 81.4% said that they had completed the vaccination schedule (Table 1).

Considering the occurrence of SLUs, we decided to group the sociodemographic information by comparing SLU+ and SLU− patients. We found that 92.3% of SLU+ patients had an income of up to the minimum wage, and 76.9% of SLU+ patients were unemployed. Overall, 88.9% of the SLU+ patients did not consume alcoholic beverages, 76.9% of patients were undergoing treatment with hydroxyurea, none of the SLU+ patients smoked, and 69.2% of SLU+ patients had completed the vaccination schedule (Table 1).

### 3.1. Social Relationships in SCD Patients with and Without SLU

Social habits and relationships were also investigated. Overall, 29.2% of the patients reported having 4–6 friends, and 25% reported having no friends. Furthermore, 45.8% of the patients reported being close to 7–10 family members. The frequency of social meetings was daily or almost every day for 29.2% of the patients, while 20.8% reported such meetings as rare. In addition, 85.4% of them did not practice sports.

Although 30.7% of SLU+ patients reported not having friends, 53.9% declared that they were close to 7–10 family members. Overall, 38.5% of SLU+ patients said that the frequency of social meetings was rare; however, 30.7% reported meeting with friends or family daily or almost every day. Notably, 84.6% of the SLU+ patients did not practice sports (Table 2).

### 3.2. Internal Consistency Reliability of the WHOQOL-BREF

The WHOQOL-BREF (Table 3) reliability assessed by Cronbach’s alpha coefficient (value = 0.88) was satisfactory considering all score items. Internal consistency was also acceptable for each score individually: the “physical health”, “psychological”, “social relationships”, and “environmental” domains (0.67, 0.84, 0.51, and 0.67, respectively).

### 3.3. Quality-of-Life Score Based on the WHOQOL-BREF Questionnaire

The WHOQOL-BREF questionnaire was used to measure the quality-of-life score among SLU+ and SLU− patients. Four domains were calculated: D1 (physical health); D2 (psychological); D3 (social relationships); and D4 (environment). D1 and D4 were higher in SLU− patients than in SLU+, while D2 and D3 were higher in SLU+ than in SLU−. We failed to find a statistically significant difference when comparing the two groups, although a numerical difference was seen. The results of all calculated domains are shown in Table 4.

### 3.4. WHOQOL-BREF Domains Are Associated with Different Features of SCD

We also tested associations between the WHOQOL-BREF domains and the characteristics of SCD patients. Statistically significant differences were observed among patients receiving different incomes; for example, higher D1 and D2 values were seen in patients receiving ≥3 times the minimum wage (Figure 1A,B). Additionally, for D4, higher values were found in patients receiving 1–2 and ≥3 times the minimum wage (Figure 1C). Likewise, increased D1, D2, and D3 values were observed in patients who were working compared to those who were unemployed (Figure 1D–F). Patients who experienced vaso-occlusion presented decreased D1 and D4 values (Figure 1G,H). In addition, female patients exhibited decreased D3 values compared to male patients (Figure 1I).

## 4. Discussion

The QoL of patients with SCD presenting SLUs can be influenced by their socioeconomic and sociodemographic profile, eating habits, and social interactions, which can impact their clinical severity. The WHOQOL-BREF assessment is an important tool for investigating health-related QoL cross-culturally and has been used in different populations, including people with SCD [1,2].

Regarding sociodemographic profiles, our patients consisted predominantly of men in the second decade of life, in accordance with other studies addressing patients with SLUs [3,4,5]. The diagnosis was most frequently observed from 6 months to 17 years, thus demonstrating a late diagnosis, which hinders preventive medical actions for clinical complications such as SLUs. In 2001, Brazil initiated newborn screening for SCD through the National Newborn Screening Program (PNTN), using the Guthrie test to detect hemoglobinopathies [6]. More recently, in 2018, the Brazilian Ministry of Health updated the clinical protocol and therapeutic guidelines for SCD, thus expanding the screening and management of the disease to mitigate the occurrence of clinical manifestations and prevent late complications, such as SLUs. [7,8].

Additionally, most patients self-declared their ethnicity as black or brown (mixed race), corroborating other studies that emphasize the relationship between ethnicity and the genetic history of the emergence of this pathology, since, according to the literature, it was through the slave trade that African individuals brought the βS allele to Brazil [9,10,11].

In terms of education, most individuals in both groups in the study had a complete middle school or partial high school education or a complete high school or partial college education. SCD patients who present SLUs usually have a high illiteracy rate [4,12], due to pain crises and the appearance and odor of ulcerated lesions. Likewise, most patients did not have an occupation when they were interviewed. Furthermore, the clinical severity of the disease prevents many patients from attending and performing work activities, and most of them are unemployed [4]. They also have a low socioeconomic level, with an income of up to the minimum wage. An observational and descriptive study evaluated patients with SLUs, finding that most had an income of one minimum wage and had difficulty attending school due to the clinical repercussions of the disease [13]. In a cross-sectional study, patients with SCD over 18 years of age had only a high school education, and most had no work activity [2]. An investigation of lifestyle habits is also necessary to explore comprehensive well-being. We found that most of the patients in both groups reported not consuming alcoholic beverages or smoking. Relationships and social habits showed that most people either had few friends or had little contact with their family. Most of the patients did not practice sports or other social activities. This is in agreement with a previous study, which also identified that patients with SLU reported having suffered from embarrassing looks, which prevented them from attending social events [13].

The results of our study confirmed the fair validity of the WHOQOL-bref as a tool for the measurement of the QoL of SLU+ and SLU− SCD patients. Our data are similar to those of the first study that tested the WHOQOL-BREF in SCD patients [2]. Despite that, all four domains calculated in our cohort were numerically lower than those found previously [2]; the sociodemographic characteristics were similar, and our data also had acceptable internal consistency, as estimated by Cronbach’s alpha coefficient. The measurement of QoL in SCD has been mainly associated with pain crises [2], chronic pain [14], and neuropathic pain [15], although different aspects, such as the impact of surgical procedures [1], have been addressed. Herein, we found that the physical and environmental domains were higher in SLU− patients than in SLU+, while the psychological and social relations domains were higher in SLU+ than SLU−, although we did not find statistical significance. We believe that a larger sample size would provide more accurate data for this analysis to assess the burden of SLUs.

Female gender, compared to male gender, was found to present a decreased social relationship domain. This finding aligns with existing literature that points to a greater vulnerability of women with SCD regarding the social and emotional dimensions of their QoL. According to previous studies, women with SCD reported significantly more frequent and severe pain as well as higher diagnosis rates of depression and anxiety than men, which can lead to a more profound impact on their social lives [3]. Additionally, difficulty in wearing certain types of clothing due to the location of the ulcers and embarrassment in public settings, such as beaches, due to the need to cover them, can contribute to a greater sense of social exclusion and stigmatization [16]. This panorama contrasts with other findings, which identified higher D3 scores for women than men, using the same methodology in Saudi patients [1]. This contrast may suggest cultural or sociodemographic variations that influence the perception and management of the social dimensions of SLUs. These factors indicate the need for specific support strategies for women with SCD and SLUs that consider the social and emotional dimensions of the condition, promoting interventions that aim to improve not only the clinical management of the disease but also the social QoL of these patients [1,3,16].

According to the literature, other studies found statistically significant differences in the four domains analyzed through the WHOQOL-BREF when comparing patients who presented 0–3 pain crises per year with those who presented >3 crises per year [2]. Earning more than three times the minimum wages is associated with a better quality of life in the physical, psychological, and environmental health domains. Likewise, patients with an occupation have a high quality of life in the physical, psychological, and social domains. Vaso-occlusive crises are associated with a worse quality of life in the physical and environmental domains.

## 5. Conclusions

Our data revealed that SCD patients’ sociodemographic characteristics regarding income, literacy, and occupation rates remain low. Income, occupation, previous episodes of vaso-occlusion, and female gender were shown to be associated with the physical and psychological domains, as well as social relationships and environment, thus affecting overall well-being. Patients with lower income and unemployment are more likely to have poorer quality of life. Although we did not find statistical differences between SLU+ and SLU− patients, utilizing the WHOQOL-BREF to evaluate QoL is necessary and feasible. Despite the limitations of our study design, we think that a better understanding of QoL in SCD patients with SLUs is required.

## Figures and Tables

**Figure 1 ijerph-22-00108-f001:**
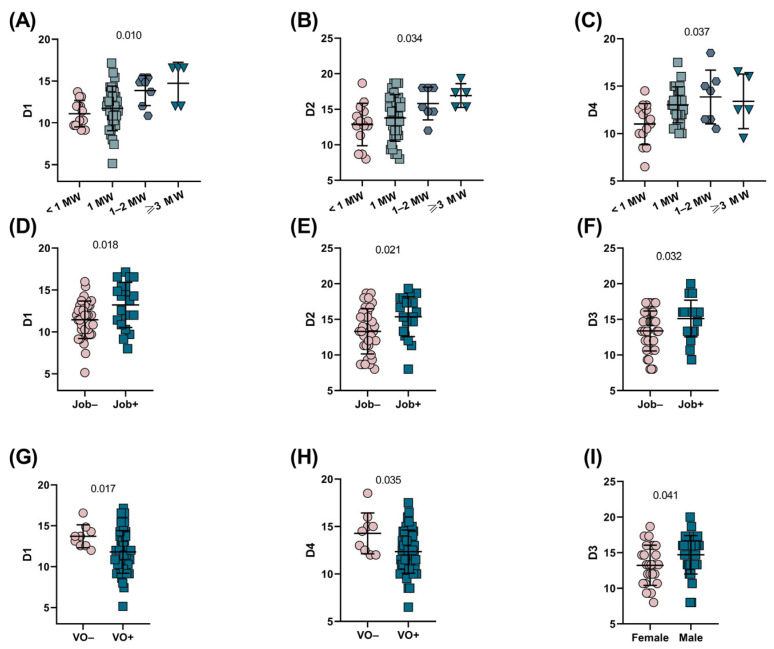
Associations between WHOQOL-BREF domains and different characteristics in SCD patients. (**A**) Patients receiving ≥3 times the minimum wage presented increased D1 values. (**B**) Patients receiving ≥3 times the minimum wage presented increased D2 values. (**C**) Patients receiving 1–2 times the minimum wage presented the greatest increase in D4 values, followed by patients receiving ≥3 times the minimum wage. *p*-values were obtained with the Kruskal–Wallis test. (**D**) Patients who had an occupation presented higher D1 values. (**E**) Patients with an occupation presented higher D2 values. (**F**) Patients who had an occupation presented higher D3 values. (**G**) Patients who experienced vaso-occlusive episodes had decreased D1 values. (**H**) Patients who experienced vaso-occlusive episodes had decreased D4 values. (**I**) Female patients exhibited decreased D3 values compared with male patients. p-value was obtained with the Mann–Whitney U test. MW: minimum wage. VO: vaso-occlusion.

**Table 1 ijerph-22-00108-t001:** Clinical, sociodemographic, and socioeconomic data on SLU+ and SLU− patients.

Characteristics	SLU+ (N = 13) N (%)	SLU− (N = 42) N (%)
**Age (years)**	33.8	34.8
**Gender**		
Female	4 (30.8)	22 (52.4)
Male	9 (69.2)	20 (47.6)
**Age at 1st SCD diagnosis**		
<6 months	1 (7.7)	7 (16.7)
6 months–4 years	5 (38.5)	6 (14.3)
5–9 years	2 (15.4)	5 (11.9)
10–14 years	1 (7.7)	4 (9.5)
15–17 years	2 (15.4)	2 (4.8)
≥18 years	2 (15.4)	18 (42.9)
**Self-declared ethnicity**		
Black	7 (53.8)	22 (52.4)
Brown	6 (46.2)	17 (40.5)
Yellow	0	1 (2.4)
Indigenous	0	2 (4.8)
**Income per capita**		
<1 minimum wage	4 (30.8)	10 (23.8)
1 minimum wage	8 (61.5)	22 (52.4)
1–2 times minimum wage	0	7 (16.7)
≥3 times minimum wages	1 (7.7)	3 (7.1)
**Education**		
Illiterate/elementary school unfinished	1 (7.7)	5 (12.5)
Elementary school completed/middle school unfinished	1 (7.7)	2 (5)
Middle school completed/high school unfinished	5 (38.5)	14 (35)
High school completed/college degree unfinished	5 (38.5)	18 (45)
Graduated	1 (7.7)	1 (2.5)
**Relatives with SCD**		
Siblings	5 (38.5)	22 (52.4)
Parents and siblings	1 (7.7)	2 (4.8)
Other relatives	2 (15.4)	6 (14.3)
None	5 (38.5)	12 (28.6)
**Hydroxyurea**		
Yes	10 (76.9)	23 (54.7)
No	3 (23.1)	19 (45.3)
**Occupation**		
Yes	3 (23.1)	17 (40.5)
No	10 (76.9)	25 (59.5)
**Consumption of alcoholic beverages**		
Yes	1 (11.1)	13 (30.9)
No	8 (88.9)	29 (69.1)
**Smoking**		
Yes	0 (0)	2 (4.8)
No	9 (100)	40 (95.2)
**Vaccination schedule completed**		
Yes	9 (69.2)	35 (85.4)
No	4 (30.8)	6 (14.6)

SLU: sickle cell leg ulcer; SCD: sickle cell disease.

**Table 2 ijerph-22-00108-t002:** Social characterization of SLU+ and SLU− SCD patients.

Social Characteristics	SLU+ (N = 13) N (%)	SLU− (N = 42) N (%)
**Friends**		
0	4 (30.7)	8 (22.9)
1	2 (15.4)	2 (5.7)
2–3	1 (7.7)	8 (22.9)
4–6	3 (23.1)	11 (31.4)
7–10	3 (23.1)	6 (17.1)
**Close to family members**		
0	1 (7.7)	2 (5.7)
1	0 (0)	0 (0)
2–3	1 (7.7)	7 (20)
4–6	4 (30.7)	11 (31.5)
7–10	7 (53.9)	15 (42.8)
**Frequency of social meetings**		
Daily or almost daily	4 (30.7)	10 (28.6)
Many times/week	1 (7.7)	6 (17.1)
Many times/month	2 (15.4)	9 (25.7)
Many times/year	1 (7.7)	5 (14.3)
Rarely	5 (38.5)	5 (14.3)
**Sports practice**		
Yes	2 (15.4)	5 (14.3)
No	11 (84.6)	30 (85.7)

SLU: sickle cell leg ulcer; SCD: sickle cell disease.

**Table 3 ijerph-22-00108-t003:** Quality-of-life scores of SCD patients using the WHOQOL-BREF questionnaire.

D1 Mean ± SD (Median, IQR)	D2 Mean ± SD (Median, IQR)	D3 Mean ± SD (Median, IQR)	D4 Mean ± SD (Median, IQR)	Cronbach’s Alpha Coefficient (All Domains)
12.11 ± 2.54 (12.00, 10.28–13.71)	14.08 ± 3.17 (14.66, 12.00–16.66)	14.02 ± 2.83 (14.66, 12.00–16.00)	12.66 ± 2.34 (12.50, 11.00–14.50)	0.88
Cronbach’s alpha coefficient of each domain				
0.67	0.84	0.51	0.67	

SD: standard deviation; IQR: interquartile range; D1: physical health; D2; psychological; D3: social relationships; D4: environment.

**Table 4 ijerph-22-00108-t004:** Quality-of-life scores of SCD, SLU+, and SLU− patients using the WHOQOL-BREF questionnaire.

Domain	SCD N = 55 Mean ± SD	SLU+ N = 13 Mean ± SD	SLU− N = 42 Mean ± SD	*p*-Value
D1	12.11 ± 2.54	11.86 ± 2.59	12.08 ± 2.49	0.758
D2	14.08 ± 3.17	14.76 ± 3.06	13.79 ± 3.20	0.404
D3	14.02 ± 2.83	14.35 ± 3.31	13.87 ± 2.71	0.554
D4	12.66 ± 2.34	12.50 ± 2.76	12.63 ± 2.20	0.706

*p*-values were obtained using the Mann–Whitney U test. D1: physical health; D2; psychological; D3: social relationships; D4: environment; SLU: sickle cell leg ulcers.

## Data Availability

Data are available upon request from the corresponding author.

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
