# Peer review of "WHOQOL-BREF in Measuring Quality of Life Among Sickle Cell Disease Patients with Leg Ulcers"

_ijerph, 2025, doi:10.3390/ijerph22010108_

Round 1

Reviewer 1 Report

Comments and Suggestions for Authors

Dear Authors,

Overall, it is an intersting paper. Sickle leg ulcers are a debilitating complication, and I consider you to bring a novelty and interesting research.

I would only suggest you to be a little bit more specific regarding the time frame set, why 2021-2022, etc., and how the control group was constructed (patients presented in the Deapartment in the same time period; similar characteristics?; why 42 patients?; etc.).

Good luck, and I hope you will get your paper published!

Author Response

For review article

Response to Reviewer X Comments

Comments 1: Dear Authors,

Overall, it is an intersting paper. Sickle leg ulcers are a debilitating complication, and I consider you to bring a novelty and interesting research.

I would only suggest you to be a little bit more specific regarding the time frame set, why 2021-2022, etc., and how the control group was constructed (patients presented in the Department in the same time period; similar characteristics?; why 42 patients?; etc.).

Good luck, and I hope you will get your paper published!

Response 1:

Thank you for pointing this out. We agree with this comment. Mention precisely where in the revised manuscript this change can be found – line 84 and lines 92-93, in the “Study design and patients” section.

Reviewer 2 Report

Comments and Suggestions for Authors

Introduction and purpose of the article

This article examines the quality of life (QoL) of patients with sickle cell disease (SCD), particularly those with leg ulcers (SLU), using the WHOQOL-BREF questionnaire. The main aim of the study was to describe the sociodemographic and clinical data of the patients and to investigate the associations between quality of life outcomes and patient characteristics.

The authors point out that the assessment of quality of life in SCD remains underdeveloped, although quality of life is a key indicator in medicine. The WHOQOL-BREF instrument was evaluated for its usefulness in measuring quality of life in this group.

Research methods

The study was a cross-sectional study involving 55 patients with SCD in Brazil. The WHOQOL-BREF questionnaire was used, which analyses four domains of quality of life:

- D1 Physical health,

- D2 Mental health,

- D3 Social relationships,

- D4 Environment.

The socio-demographic and clinical data were collected by inspecting the medical records and interviewing the patients. The statistical analysis was carried out using the Shapiro-Wilk test, the Mann-Whitney U test and Cronbach's alpha coefficient.

Results

Patient profile:

- The mean age was 34.9 years, and the majority of patients had low income (less than minimum wage) and high unemployment (63.6%).

- SLU patients were less likely to be employed and had a higher proportion of incomplete education than the control group (SLU-).

Quality of life:

- Scores in the quality of life domains varied. SLU- patients had higher scores in the areas of physical health (D1) and environment (D4), while SLU+ patients had better scores in the areas of psychological (D2) and social relationships (D3).

- There were no statistically significant differences between the SLU+ and SLU- groups.

Factors influencing quality of life:

- Higher income and employment were associated with better physical, psychological and environmental outcomes.

- Vascular and thrombotic crises and female gender were associated with lower QoL scores.

Discussion

The article emphasises the important influence of socioeconomic factors, such as income and employment, on the quality of life of patients with SCD. The authors note that the results are consistent with previous studies showing that people with a lower socioeconomic status have a poorer quality of  life.

One interesting finding was that women had lower scores than men in the social relationships domain (D3), which may be related to women's greater emotional sensitivity to the stigma associated with SCD and SLU.

The authors suggest that the lack of significant differences between the SLU+ and SLU- groups may be due to the limited sample size and recommend further research with a larger number of participants.

Strengths of the article

- Usefulness of the WHOQOL-BREF: The study confirms that the WHOQOL-BREF is an appropriate tool to assess quality of life in this patient group.

- Comprehensive approach: Consideration of a wide range of sociodemographic, clinical and quality of life data.

- Statistical reliability: The use of different statistical methods increases the reliability of the results.

Weaknesses of the article

1 Limited sample: The sample size (55 patients) was too small to achieve statistically significant results in some analyses.

2. Lack of geographic diversity: The study is limited to one region in Brazil, which may limit the generalizability of the results to other populations.

However, further studies with a larger number of participants and a more detailed clinical approach are needed to better understand these relationships.

Recommendations for authors

1 Include detailed information about patients' therapies, such as drug treatment or psychosocial interventions.

2. Complete the details of the therapies: No details of patients' therapies were provided, which may have affected quality of  life.

In follow-up studies:

1. Consider increasing the study sample size to obtain more robust results.

2. Extend the study to other regions or countries to see if the results are consistent in different cultural contexts.

Author Response

For review article

Response to Reviewer X Comments

Comments 2: Introduction and purpose of the article

This article examines the quality of life (QoL) of patients with sickle cell disease (SCD), particularly those with leg ulcers (SLU), using the WHOQOL-BREF questionnaire. The main aim of the study was to describe the sociodemographic and clinical data of the patients and to investigate the associations between quality of life outcomes and patient characteristics.

The authors point out that the assessment of quality of life in SCD remains underdeveloped, although quality of life is a key indicator in medicine. The WHOQOL-BREF instrument was evaluated for its usefulness in measuring quality of life in this group.

Research methods

The study was a cross-sectional study involving 55 patients with SCD in Brazil. The WHOQOL-BREF questionnaire was used, which analyses four domains of quality of life:

- D1 Physical health,

- D2 Mental health,

- D3 Social relationships,

- D4 Environment.

The socio-demographic and clinical data were collected by inspecting the medical records and interviewing the patients. The statistical analysis was carried out using the Shapiro-Wilk test, the Mann-Whitney U test and Cronbach's alpha coefficient.

Results

Patient profile:

- The mean age was 34.9 years, and the majority of patients had low income (less than minimum wage) and high unemployment (63.6%).

- SLU patients were less likely to be employed and had a higher proportion of incomplete education than the control group (SLU-).

Quality of life:

- Scores in the quality of life domains varied. SLU- patients had higher scores in the areas of physical health (D1) and environment (D4), while SLU+ patients had better scores in the areas of psychological (D2) and social relationships (D3).

- There were no statistically significant differences between the SLU+ and SLU- groups.

Factors influencing quality of life:

- Higher income and employment were associated with better physical, psychological and environmental outcomes.

- Vascular and thrombotic crises and female gender were associated with lower QoL scores.

Discussion

The article emphasises the important influence of socioeconomic factors, such as income and employment, on the quality of life of patients with SCD. The authors note that the results are consistent with previous studies showing that people with a lower socioeconomic status have a poorer quality of  life.

One interesting finding was that women had lower scores than men in the social relationships domain (D3), which may be related to women's greater emotional sensitivity to the stigma associated with SCD and SLU.

The authors suggest that the lack of significant differences between the SLU+ and SLU- groups may be due to the limited sample size and recommend further research with a larger number of participants.

Strengths of the article

- Usefulness of the WHOQOL-BREF: The study confirms that the WHOQOL-BREF is an appropriate tool to assess quality of life in this patient group.

- Comprehensive approach: Consideration of a wide range of sociodemographic, clinical and quality of life data.

- Statistical reliability: The use of different statistical methods increases the reliability of the results.

Weaknesses of the article

1 Limited sample: The sample size (55 patients) was too small to achieve statistically significant results in some analyses.

  1. Lack of geographic diversity: The study is limited to one region in Brazil, which may limit the generalizability of the results to other populations.

However, further studies with a larger number of participants and a more detailed clinical approach are needed to better understand these relationships.

Recommendations for authors

1 Include detailed information about patients' therapies, such as drug treatment or psychosocial interventions.

  1. Complete the details of the therapies: No details of patients' therapies were provided, which may have affected quality of  life.

In follow-up studies:

  1. Consider increasing the study sample size to obtain more robust results.
  2. Extend the study to other regions or countries to see if the results are consistent in different cultural contexts.

Response 2:

Thank you for pointing this out. We agree with this comment. Mention precisely where in the revised manuscript this change can be found – lines 112-118, in the "Clinical, sociodemographic, and socioeconomic data" section.

We agree that increasing the sample size would provide more robust results. However, as reported in other studies (DOI: 10.1371/journal.pone.0274254; DOI: 10.1371/journal.pone.0186270) this is very difficult due to the profile of patients with leg ulcers. We believe that a longer period of analysis and the addition of more study centers in different regions could reach more participating patients.
